# COVID-19 Effects on Environmentally Responsible Behavior: A Social Impact Perspective from Latin American Countries

**DOI:** 10.3390/ijerph20043330

**Published:** 2023-02-14

**Authors:** Leslier Valenzuela-Fernández, Manuel Escobar-Farfán, Mauricio Guerra-Velásquez, Elizabeth Emperatriz García-Salirrosas

**Affiliations:** 1Department of Administration, School of Economics and Business, University of Chile, Santiago 8330015, Chile; 2Department of Administration, Faculty of Administration and Economics, University of Santiago of Chile, Santiago 9170020, Chile; 3Faculty of Engineering and Management, Professional School of Business Administration, Universidad Nacional Tecnológica de Lima Sur, Lima 15816, Peru; 4Faculty of Management Sciences, Universidad Autónoma del Perú, Lima 15842, Peru

**Keywords:** environmental behavior, purchase intention, theory of planned behavior, environmental consciousness, COVID-19

## Abstract

The aim of the research is to examine the relationships between the following variables (a) Theory of Planned Behavior (TPB), composed of Attitudes (ATT), Subjective Norms (SN), and Perceived Behavioral Control (PBC); and (b) Consciousness (EC) on the dependent variable Environmentally Responsible Purchase Intention (ERPI) from the perspective of the Latin American consumer in a pandemic. Currently, the literature on the relationships proposed in the explanatory model is still scarce at a theoretical and practical level, without empirical evidence in Latin America. The data collection is from 1624 voluntary responses from consumers in Chile (n = 400), Colombia (n = 421), Mexico (n = 401), and Peru (n = 402) collected through online surveys. Using structural equation modeling (SEM) and multi-group to test invariance analysis and the moderating effects, we can determine the relationship between the variables in the proposed model, generating evidence from Latin American countries. The empirical analysis verified that Attitude (ATT), Perceived Behavioral Control (PBC), and Environmental Consciousness (CE) have a positive and significant effect on Environmentally Responsible Purchase Intention (ERPI). The results also show that the generation variable presents invariance. Therefore, the groups are not different at the model level for the generation variable, becoming relevant to the difference at the path level. Therefore, the results of this study become a relevant contribution, indicating a moderating effect on the generation variable. This research provides insights for understanding Latin American consumers, and managerial implications are also provided for developing strategies to promote sustainable consumption.

## 1. Introduction

Currently, society has shown a massive interest in Environmental Consciousness (CE), where industries are concerned with mitigating environmental damage resulting from their economic activities [1,2,3,4]. Since December 2019, the international health crisis related to COVID-19 allowed the evaluation of environmental impacts in different contexts. In this vein, COVID-19 has deeply affected economic structures, as well as social and commercial relations. Particularly, the pandemic has been a phenomenon that has generated positive and negative effects on the global environment [4,5,6,7,8,9]. Restrictions to the free movement between and through the region’s urban areas decreased economic activity as well as the use of cars, trucks, and other motorized vehicles. As a result, many cities in Latin America experienced a short-run air pollution reduction. For example, Bogotá showed the most significant decrease (−83%), and Guayaquil, Rio de Janeiro, and São Paulo were the cities with the lowest reduction of 30% [10], places that historically had problems with environmental pollution. In this sense, the pandemic and lockdown measures have temporarily reduced urban pollution in many Latin American cities. Additionally, COVID-19 has positively decreased water and noise pollution due to confinement in homes and travel restrictions [11], improving the efficiency of the consumption of a natural resource in permanent monitoring for its potential shortages in the coming years. An additional positive consequence of this pandemic is the generation of a collective awareness regarding protecting the environment and the need to conserve natural resources [12]. Indeed, negative environmental contexts positively affect the responsible environmental behavior of the population, which has been observed in the context of COVID-19 [13].

On the other side, we face a consumer society of manufactured products and services that negatively affect the environment [14,15]. This situation was aggravated with the arrival of COVID-19, where individuals around the world have adopted an excessive use of disinfectants and disposable materials such as face shields, masks, bags, and other products related to the prevention of the spread of the virus that caused a high amount of plastic and sanitary waste [16].

Environmentally responsible behavior can facilitate improving the environment and advancing a sustainable society [17,18,19]. In this context, emergent research findings have linked the effects of COVID-19 on consumers’ perceptions, environmental awareness, or/and purchase intention [20,21,22,23]. However, the literature on the relationships proposed in the explanatory model is still scarce at a theoretical and practical level, with no empirical evidence in Latin America.

Based on the above, the experience of this world crisis could lead to changes in behavior at the collective and individual levels. Therefore, there is a research gap to empirically evaluate if this is observed in different contexts, especially in Latin American consumers, since there is very little evidence exploring this phenomenon in this geographical area. The importance of a better understanding of the potential effects of COVID-19 on environmentally responsible behavior is the insights that new evidence can provide for designing behavior-based government and business policy instruments that focus on changing consumption and production patterns [24].

The Theory of Planned Behavior (TPB) is one of the most relevant social psychology theories; it explains individual behavioral intentions [25] and is the theoretical approach applied in this research. This theory was built based on the seminal work of Fishbein and Ajzen [26] and the Theory of Reasoned Action (TRA), which explains human actions as a consequence of behavioral intentions that are influenced by attitudes and perceived social pressure. Later, in 1985, Ajzen argued that in addition to the Theory of Reasoned Action, when it comes to human behavior, it is necessary to consider the perception of perceived control of internal or external factors that influence determined behavior (locus of control) [27]. According to Ajzen [28], three main factors in TPB influence individual behavior. First, attitudes refer to the degree to which a person values, positively or negatively, a behavior [27,29]. Second, Subjective Norms, are the social pressure or normative expectations of social groups to which individuals belong. Third, Perceived Behavioral Control (PBC) is a predictor that reflects the ease or difficulty of the individual’s perception of performing a particular behavior [28,30]. In this sense, this variable is analyzed from self-efficacy, which corresponds to the individual’s degree of effort to conduct a potential behavior [30,31]. Therefore, the three determinants of this theoretical approach can be observed at individual and social levels. While in 1992, it was argued that TPB is a better theoretical framework than TRA to predict human behavior [27], recent research suggests and recommends the application of TPB as a broader and more appropriate predictor to understand consumer behavior [32,33,34].

The main purpose of this study is to contribute to the theoretical and empirical gap about the effects of COVID-19 on environmentally responsible behavior in Latin American countries from a TPB perspective. Therefore, the main contributions of this article are (1) to understand and corroborate the effect of COVID-19 on Latin American consumers’ declaration of Environmentally Responsible Purchase Intention (ERPI); (2) to test the validity, reliability, and statistical significance of each hypothesis of the explanatory model by using structural equation modeling (SEM); (3) to examine the estimation performance of the extended TPB model for Latin American consumers; (4) to test moderating effect and test invariance for the hypothesis of the model by using multi-group of generations. 

Thus, to fill this gap, this study responds to the lack of research to examine the relationship between the following dimensions a) Theory of Planned Behavior (TPB), composed of Attitudes (ATT), Subjective Norms (SN), and Perceived Behavioral Control (PBC); and b) Environmental Consciousness (CE), on the dependent variable of Environmentally Responsible Purchase Intention (ERPI) from the perspective of the Latin American consumer in a pandemic context. Building upon Severo et al. [4] and Xu et al. [23], this study presents a quantitative statistical analysis employing Structure Equation Modelling (SEM).

The rest of the article is structured as follows: Section 2 provides the conceptual background and hypotheses of this research. Section 3 details the research methodology. Section 4 presents the results. Section 5 offers a discussion of our evidence based on theoretical and managerial implications identified in the field. Finally, the article presents the main conclusions and identifies future research avenues. 

## 2. Theoretical Background

Following the objectives of this research and the context in which it was carried out, the literature review is based on the formative model proposed by Kumar et al. [15] and the hypothesis proposed in the context of the COVID-19 pandemic (See Figure 1). Below is a theoretical framework that supports the constructs that are the basis of our hypotheses and theoretical model:

### 2.1. COVID-19 and Environmentally Responsible Purchase Intention

The intention has to do with the willingness to adopt a particular behavior, and purchase intention is a prerequisite to stimulate and push consumers to buy products and services [35,36]. Consumers’ concern about the environment affects their purchasing decisions [35,37]. Nowadays, consumers deciding on the type of product to buy requires being informed about both the environmental situation and the benefits responsible consumption brings to its conservation [38]. That is, green purchase intention is directly and strongly influenced by perceived value, attitude, and trust [39], whereas perceived behavioral control, perceived consumer effectiveness, subjective norm, perceived green quality, and environmental concern are moderately related to green purchase intention [35]. In previous research, environmentally responsible purchase intention (ERPI) has been extensively studied, taking into account intention in the Theory of Planned Behavior (TPB), which is made up of three elements: attitude, subjective norm, and perceived behavioral control [40,41,42,43].

With the arrival of the COVID-19 pandemic, several studies have explored its influence on consumers’ sustainable consumption behavior [3,18,25,44,45,46]. For instance, Severo et al. [4] found that the COVID-19 pandemic is essential in changing people’s behavior, affecting environmental sustainability and social responsibility. Notably, the impact of the COVID-19 pandemic had a more significant impact on sustainable consumption fluency, followed by environmental awareness and, to a lesser extent, social responsibility. Valenzuela et al. [3] analyzed the impact of the COVID-19 pandemic on consumer behavior in four Latin American countries and found that consumers reported that their behavior had become more ecologically and socially responsible. In addition, these respondents indicated that they had increased their interest in sustainable consumption and purchasing environmentally friendly products to reduce waste and negative environmental impacts. For their part, Dayanto et al. [47] found that people’s beliefs that the pandemic was caused by humanity’s excessive intrusion into nature have a positive impact on their environmental awareness. This, in turn, demonstrates a positive behavioral change toward the environment. Another study by Dangelico et al. [45] found that COVID-19 generated relevant changes in consumer behavior. Those who have increased their purchase frequency and willingness to pay for sustainable products show increasing attention to environmental issues and behave more sustainably. 

Furthermore, the extent of change is strongly affected by sociodemographic variables such as gender, age, income, and education. For example, women reported a greater shift towards sustainable consumption and behavior than men. Brzustewicz and Singh [44] found six relevant themes related to sustainable consumption: (1) organic food consumption, (2) food waste, (3) vegan food, (4) sustainable tourism, (5) sustainable transportation, and (6) sustainable energy consumption. They also identified four clusters: lifestyle and climate change, responsible consumption, energy consumption, and renewable energy.

The COVID-19 pandemic has not only impacted individual consumption behavior; some studies indicate that industrial consumption has also been affected. For instance, the study by Ruttis [46] highlights the positive impact of COVID-19 on the raw material production, raw material processing, and packaging manufacturing stages of the value chain, as well as the neutral impact in the product manufacturing stage and the negative impact in the retail stage. All this background suggests that the COVID-19 pandemic has impacted the consumers’ consumption decisions and, thus, purchase intent, making it more conscious and focused on sustainability.

### 2.2. Conceptual Model Development and Research Hypothesis in the Context of the COVID-19 Pandemic

The literature review is based on the formative model development of this research [48] and the proposed hypothesis in the context of the COVID-19 pandemic (See Figure 1). In the last decades, the Theory of Planned Behavior (TPB) has been considered one of the most reliable and accurate for analyzing environmental behavior in individuals [23,49]. Its importance can be observed in the broad range of disciplines and areas that have applied this theoretical approach, such as psychology, business and management, education, public health, and environmental sciences [50]. For example, this model has been applied in different contexts of green consumer or eco-friendly consumption [15,23,51,52]. In this sense, this theory argues that human behavior is guided by three main considerations: beliefs about the potential experiences and consequences related to behaviors (behavioral beliefs) that produce unfavorable and favorable attitudes toward specific behaviors, beliefs about the behaviors of significant others, and the normative expectations that its produce (normative beliefs), manifested as a perception of social pressure or a subjective norm, and beliefs related to the presence of factors that could facilitate or hinder the performance of the behavior (control beliefs), linked to the perceived behavioral control or self-efficacy on individuals [53].

In the area of consumption as a planned behavior, previous research has shown that it is possible to examine purchase intention through the dimensions of TPB [20,21,22,23]. However, Kumar et al. [15] have pointed out that it is possible to improve the predictability of behavior by including a new variable: Environmental Consciousness (EC). Based on the above, Figure 1 illustrates the conceptual framework applied in this research. Each element of the framework is discussed as follows:

#### 2.2.1. Attitude and Environmentally Responsible Purchase Intention

In the emergence of attitudes, behavioral beliefs play a central role by linking outcomes and experiences with the behavior of interest. In this sense, these beliefs act as a subjective evaluation of the potential outcome’s experiences by specific behaviors, which are mixed with subjective values of the expected results, determining the attitude toward behavior. Therefore, when there is a more favorable attitude towards a specific performance, the intention to act is more probable [30,54]. According to Ajzen [29], intention indicates a person’s readiness to perform a given behavior. Therefore, it can be assumed as an immediate antecedent of behavior.

In this regard, evidence in the literature shows that individuals with a positive attitude toward the environment are more willing to perform pro-environmental activities and behaviors [55,56,57]. Furthermore, a positive relationship between becoming aware of the environment’s protection and manifesting an attitude toward its care has been reported [23,58,59]. A positive attitude generates a good feeling in people because they contribute to caring for the environment and protecting nature [20,23,60]. In other words, environmental impacts related to intentions to buy pro-environmental products are mediated by attitudes [61].

Scholars have argued that the COVID-19 pandemic has contributed to a positive attitude toward responsible purchasing [62]. For instance, Peluso et al. [63] suggested that the intention to consume sustainable products has increased. Meanwhile, Alexa et al. [64] have noted that consumers have maintained their attitude toward buying sustainable products from local brands. However, according to Wang and Li [65], if individuals have a favorable attitude toward consuming when they believe they will achieve their desires, they will act with a happy readiness to consume. Accordingly, to the previous discussion, the following hypothesis is suggested:

**H1.** *Attitude (ATT) determined the consumers’ environmentally responsible purchase intention (ERPI) during COVID-19*.

#### 2.2.2. Subjective Norms and Environmentally Responsible Purchase Intention

According to Chiu et al. [32], close people should influence an individual’s behavior, such as family, friends, neighbors, or co-workers [30,34,64]. Previous studies have argued that the Subjective Norm is a relevant factor in individuals’ efforts when purchasing environmentally friendly products or services [40,66,67].

Hence, the intention of sustainable purchases with the environment can decrease or increase depending on social pressure [56]. In the context of COVID-19, some authors have suggested that Subjective Norms have been affected by the social pressure of this global health crisis, increasing the intention of environmentally responsible behavior [62,68]. Furthermore, there is evidence that Subjective Norms affect purchasing intention during the COVID-19 pandemic through mobile apps [69]. For example, Zebardast and Radaei [20] reported that the pandemic had raised consumers’ knowledge of their environments, affecting individuals’ Subjective Norms an improving pro-environmental social behavior. Based on this discussion, we propose the following hypothesis:

**H2.** *Subjective Norms (SN) determined the consumers’ environmentally responsible purchase intention (ERPI) during COVID-19*.

#### 2.2.3. Perceived Behavioral Control and Environmentally Responsible Purchase Intention

PBC has been related to behavioral purchase intention [34,70] by influencing perceptions and control beliefs [33]. Xu et al. [23] have suggested that PBC considers consumers’ disposition concerning time, money, and ability in the context of green purchase intention. Recent evidence indicates that consumers define PBC as the ease or difficulty of behaving in an environmentally friendly approach [56]. Nonetheless, little research has empirically examined this dimension in the context of responsible purchase intention during COVID-19 [25]. In European countries, the relationship between the pandemic on Perceived Behavioral Control shows a positive behavior toward purchasing sustainable products [64]. Additionally, Zebardast and Radaei [20] state that Perceived Behavioral Control during the COVID-19 pandemic has positively influenced consumers since they have increased their relationship between intention and environmentally responsible behavior. In other words, people are concerned about environmental behavior, such as climate change, increasing their awareness regarding these issues during pandemic times [71]. Based on the above, the following hypothesis is proposed:

**H3.** *Perceived Behavioral Control (PBC) determined the consumers’ environmentally responsible purchase intention (ERPI) during COVID-19*.

#### 2.2.4. Environmental Consciousness and Environmentally Responsible Purchase Intention

Environmental Consciousness (EC) is commonly defined as the orientation toward consuming environmentally friendly goods, which causes people to think about more ecological decision-making [67,72,73,74]. According to Bamberg [75], the initial thoughts with emphasis on environmental concerns were the critical elements for individuals to perform environmentally friendly behaviors. As a result, people change their way of acting to contribute to resolving ecological problems [74,76,77]. In this vein, Kumar et al. [15] argued that consumers are concerned about making purchases to protect the environment. Moreover, when individuals manifest Environmental Consciousness, it increases their purchase intention of eco-friendly products [78]. It has also been argued the importance of making products and brands with a visible positive impact on the social challenges that society is facing affecting EC. In this sense, the Purchase may not be concluded if there is no clear stance reflecting responsibility to the environment [79,80].

Every day there is greater awareness of the damage to the planet and the environment, and the pandemic has accelerated community responsibility and EC [81]. Based on the recent evidence, we propose the following hypothesis:

**H4.** *Environmental Consciousness (EC) determined the consumers’ environmentally responsible purchase intention (ERPI) during COVID-19*.

### 2.3. Moderating Effect

This study has included the moderation analysis to observe how the age of individuals could influence the effect of Attitude, Subjective Norms, Perceived Behavioral Control, and Environmental Consciousness with Environmentally Responsible Purchase Intention during COVID-19. According to Severo et al. [4], cultural and socioeconomic contexts must be assessed since these factors may influence the intensities of people. For this reason, the study presupposes the moderation effect was evaluated according to the respondents’ date of birth, grouped by generation. The cohort by years between one generation and another varies between authors. Still, we use the following order: Generation X from 1965 to 1977, Generation Y or Millennials from 1977 to 1994, and Generation Z or Centennials from 1995 to 2012 [4,82]. A study developed by Deloitte [83] found that the biggest concern of millennials was climate change, protecting the environment, and natural disasters, and the findings of the Naderi and Van Steenburg [84] study, in general, reveal that rational and self-oriented rather than emotional and other-oriented motives lead millennials to act pro-environment at work.

Furthermore, different studies have analyzed the relationship between some sociodemographic or personal characteristics and green consumption. For example, a study by Wong et al. [85] states that people over 50 years old tend to accept products with low carbon emissions. Likewise, the results indicate a significant interaction between ecological attractiveness and age groups. Therefore, by adding an environmental attraction, the growth in the declared preference is more intense in the older groups (over 40 years old) than in the younger ones (between 20 and 30 years old) [86].

On the other hand, some studies show differences between the age group regarding social norms, attitudes, and environmental awareness. For example, some results of the casual effect show that green defenders in the age group older than 50 tend to accept easily low-carbon products more than two other age groups. On the contrary, the influence of social norms on consumers in the 50s age group is comparatively less significant than in other younger groups.

These findings fit the TPB [29] and the results of the study by Trivedi et al. [87] in which social norms influence the development of environmental awareness in society.

Based on this argument, we present the following hypothesis:

**H5a.** *Generations moderate the relationship between Attitude (ATT) and Environmentally Responsible Purchase Intention (ERPI)*.

**H5b.** *Generations moderate the relationship between Subjective Norms (SN) and Environmentally Responsible Purchase Intention (ERPI)*.

**H5c.** *Generations moderate the relationship between Perceived Behavioral Control (PBC) and Environmentally Responsible Purchase Intention (ERPI)*.

**H5d.** *Generations moderate the relationship between Environmental Consciousness (EC) and Environmentally Responsible Purchase Intention (ERPI)*.

## 3. Methodology

### 3.1. Context and Method

This article aims to examine and understand the effects of COVID-19 on environmentally responsible purchase intention. More specifically, this research focused on how COVID-19 has influenced Attitudes, Subjective Norms, Perceived Behavioral Control, and Environmental Consciousness. This work was carried out from consumers’ perspectives in Latin American countries through a quantitative method. The sample data included consumers from Chile, Colombia, Mexico, and Peru.

For this study, the global explanatory model is composed of formative constructs and reflective indicators. That is, the independent variables are formative of the dependent variable, and the indicators representing each construct or variable are reflective. After defining the formative or reflective nature of the indicators of the constructs or variables established in the model, we evaluate the measurement model.

In this study, a cross-sectional research method was applied through a self-administered questionnaire [88]. The selected instrument was proposed through pre-established and validated scales [15]. The survey was applied in Spanish-speaking countries. Therefore, it was necessary to use the back-translation method proposed by Brislin [89].

### 3.2. Sample and Procedure

COVID-19 has caused different confinement and social distancing restrictions in Latin American countries [90]. For this reason, this research utilized an online survey method through the Google form platform. The survey was applied to local people in Chile, Colombia, Peru, and Mexico for three months (July to September 2021). In addition, the authors decided to share the questionnaire on the internet through social networks, such as Facebook, Instagram, LinkedIn, e-mail, and WhatsApp.

Since the literature on the relationships proposed in the explanatory model is still scarce at a theoretical and practical level, with no empirical evidence in Latin America, this study is exploratory [91]. A non-probabilistic and convenience data sampling method was applied to collect data [92]. Furthermore, the snowball sampling method was used through social networks. This technique makes it possible to expand the geographical location and reduce the respondents’ access barriers [93,94]. This research is ad hoc to apply this method to minimize respondents’ risk of COVID-19. Moreover, social networks and the internet contribute to the non-probabilistic aspect of random and diverse respondents [95].

Respondents were informed that the data collected would be used exclusively for academic purposes, and the data were analyzed anonymously. Consequently, sociodemographic information was analyzed to understand the study sample. This procedure resulted in a sample size of 1624 consumers who responded to the survey in four countries. The sample per country was 24.6% from Chile (n = 400), 25.9% from Colombia (n = 421), 24.7% from Mexico (n = 401), and 24.8% from Peru (n = 402). The sample is equitable between each country; therefore, the research sample size meets the required requirements [96].

The sample data showed most of the participants were women. There were 943 female participants (58.1%), followed by 670 male participants (41.3%), and 7 participants (0.6%) of the respondents decided not to reveal their gender. In terms of age, the sample was categorized by groups of age, generation X (11.7%, n = 190), generation Y (millennials) (42.9%, n = 697), and generation Z (45.4%, n = 737).

Since the present work is an exploratory study [91], consistent with the type of non-probability sampling [97], it is not possible to calculate sampling errors. However, being the smallest sample size of 197 and when analyzing the composite reliabilities of the whole sample, the values in all factors were high (>0.90). This indicates that the items adequately measure the intended factor, which is why the literature considers it appropriate to work with small samples [92].

After the study was carried out, to verify that the differences in the subsample sizes concerning the generation did not constitute a problem, the power analysis was performed using the independent samples *t*-test (given that they are different generations), using SPSS-27. This was determined using the smallest sample (197) and the most significant sample (737) because they are the most distant and to ensure greater reliability in these results. The power analysis shows a value of 0.980, which is above 0.80 of the minimums required, which allows us to affirm that there is no problem with distant sample size differences [98] (See Table 1).

The final sample collection was classified into five groups according to the total monthly income. First, the sample reveals a higher percentage of participating people who declare earning at least two minimum salaries in their country (36.8%, n = 598). Then, followed by three and four average minimum salaries (25.4%, n = 413), a third group that has declared between five and ten average minimum salaries (23.6%, n = 376), and then a group has from eleven to twenty minimum average salaries (9.5%, n = 155), and finally a small group over twenty minimum salaries (5.0%, n = 82).

### 3.3. Measures

To construct this model research, the items were adopted from previous studies. Eight academics from different areas, such as marketing, management, and sustainability, tested and evaluated the questionnaire to check the scales and items.

The final questionnaire was divided into two parts. The first section presented 18 questions related to the COVID-19 effects on environmentally responsible purchase intention. Specifically, this section was separated into six dimensions with three-item each.

The four independent variables used in our model were (i) Attitude (ATT) [9,17], (ii) Subjective Norm (SN) [9,67], (iii) Perceived Behavioral Control (PBC) [9,74], and (iv) Environmental Consciousness (EC) [9,75,76], and the dependent variable was (v) Environmental Responsibly Purchase Intention (ERPI) [9]. Finally, the moderating variable was age, using generations X, Y, and Z.

The second section was related to six questions about demographic data, such as country, age, gender, income, educational level, and civil status.

Every item was written as a statement to be evaluated (See Table 2), applying a five-point Likert scale ranging from 1 (totally disagree) to 5 (totally agree). In this sense, all the participants interviewed were explained every number and item to understand and respond adequately [99].

### 3.4. Statistical Analysis

To establish the identification of the proposed explanatory model, we verified that the two most basic heuristic rules are fulfilled. In this sense, all the constructs have at least three indicators, and it is a recursive model.

The first statistical process evaluated the reliability and validity model. Specifically, this study measured the reliability of latent variables and the internal consistency of the items using the Cronbach Alpha method. Then, a confirmatory factor analysis (CFA) was applied to verify the fit and measurement model. The research employed two statistical software. First, the IBM SPSS Statistics software was used to check the convergent and discriminant validity. Secondly, the AMOS Software was selected to test and propose the model and hypothesis through a multi-group Structure Equation Modelling (SEM).

SEM is considered a suitable method for this type of research. First, this method is highly recommended to evaluate cause–effect relations in descriptive models [92]. Secondly, SEM is a perfect method to test the hypothesis of dependence relationships, correlations, and effects of moderating variables [104]). Finally, recent studies applied SEM to analyze and demonstrate robustness in measures and structural assessment [15,105,106].

## 4. Results

In the following section, we present the results of Reliability analysis and convergent and discriminant validity, the SEM estimates of the general proposed hypotheses; therefore, for this, the calculations were performed for the entire sample jointly, that is, for the total sample of 1624; in addition, the analysis of invariance and moderating effects was performed, for which the subsamples by groups were used both by country and by generation.

### 4.1. Reliability and Validity Analysis

Cronbach’s alpha [107] is the most widely used reliability indicator in construct verification scales [48,108]. A level considered adequate is a value of the latent variables greater than 0.70 [92,108]. Our results are satisfactory since each indicator is more significant than 0.94 (See Table 3). Regarding Convergent validity, the mean extracted variance (AVE) and Composition Reliability (CR) are utilized. AVE indicator is considered acceptable with values equal to or greater than 0.5 [92,108]. CR should be greater than 0.6 [109]. In this research, each latent variable shows a good level, AVE with values greater than 0.766 and CR with values equal to or higher than 0.908. Additionally, it is corroborated by the Correlation Matrix (See Appendix A).

In the same vein, this study incorporates the Fornell Larcker criteria [110] and the heterotrait–monotrait ratio of correlations for discriminant validity [111]. Again, the results establish an adequate validity of the model proposed (See Table 4 and Table 5).

In addition, the questionnaire was also verified. This procedure is essential to exclude the possibility of common technique bias (CMB). Similarly, Common method variance (CMV), an approach utilizing valued marker variables, was used to examine the CMV. To employ this analysis on the environmental framework, a non-ideal marker (having indirect empirical related) was utilized as the marker variable. The analysis compares different CFA models with the marker variable. The method-C (constrained model) fits significantly better than the baseline model (evidence of shared CMV between the indicators of substantive variables and the latent marker variable). The method-U (unconstrained model) does not fit considerably better than Method-C; all indicators have the same CMV. In conclusion, the method-R model is not statistically significant (LR: 9.52, *ρ* = 0.484) when compared to the method-C model (LR: 184, *ρ* = 0.000000000) or method-U model (LR: 59.3, *ρ* = 0.00002834), demonstrating that the presence of CMV has to effect on the link between the substantive variables. Consequently, the data indicate that CMV is insufficient to cause a bias in the outcomes (See Table 6).

### 4.2. SEM Estimations of the General Proposed Hypothesis

The results have indicated that the measurement model provided good model fit values. Specifically, with $\chi^{2}/df$ = 4.693 (375.444/80), an acceptable value is between 3 and 5 [96]. In the case of incremental values, to be considered a good fit with values: NFI (Normed Fit Index) > 0.90 [112], TLI (Tucker–Lewis Index) > 0.90 [113], and CFI (Comparative fit Index) > 0.95 [114]. The absolute index is acceptable when GFI (Goodness of Fit Index) is greater than 0.90 [113]. Finally, the parsimony indexes are good model fit whit values; RMSEA (Root Mean Square Error of Approximation) from ≤0.05 to 0.08 [114] and PGFI (Parsimony Goodness-of-Fit) > 0.5 [115]. The results have indicated a satisfactory Goodness of Fit, CFI = 0.958, NFI = 0.984; TLI = 0.983; RMSEA = 0.055; GFI = 0.958; PGFI = 0.639.

According to the influence of COVID-19 on environmentally responsible purchase intention (ERPI), the study reveals a positive and significant relationship in three dimensions. Attitude (ATT → ERPI; CR = 4.043 ***; SE = 0.029); Perceived Behavioral Control (PBC → ERPI; CR = 3.921 *** SE = 0.038), and Environmental Consciousness (EC → ERPI; CR = 20.099 *** SE = 0.033). However, the hypothesis related to Subjective Norms (SN → ERPI; CR = 1.708) SE = 0.034 is not significant. Consequently, hypotheses H1, H3, and H4 are supported in this research (See Table 7 and Figure 2).

### 4.3. Invariance Analysis and Moderating Effects: Generation

An invariance analysis and moderation effects were carried out to observe differences in generational groups (See Table 8 and Table 9). Three groups are selected to analyze the role of age as a moderating variable for the measurement model generation X (11.7%, n = 190), generation Y (millennials) (42.9%, n = 697), and generation Z (45.4%, n = 737). SEM is applied using the maximum-likelihood method in consideration of the model structure. The invariance analysis of the measurement model is used to determine the difference at the model level or path level [116].

The results show that the groups are not different at the model level; these groups show differences at the path level (see Table 8). Age plays a moderator role, which exposes the relevance of Environmentally Responsible Purchase Intention (ERPI) in generation X, and Perceived Behavioral Control (PBC) influence in the younger group (generations Y and Z).

## 5. Discussion and Conclusions

This research was conducted to empirically evaluate different factors that can affect the Environmentally Responsible Purchase Intention of consumers in Colombia, Mexico, Chile, and Peru in the context of COVID-19. The global explanatory model is composed of formative constructs and reflective indicators. This explanatory model is composed of five independent variables that collectively explain a phenomenon related to the dependent variable, Environmental Responsibly Purchase intention [117,118,119,120]. The results show evidence that supports the hypothesis proposed in prior studies concerning COVID-19’s influence as a phenomenon that has affected the population regarding environmental issues in society [1,4]. In terms of research findings (see Table 7), our main theoretical contribution is that our evidence supports the idea that Attitude (ATT), Perceived Behavioral Control (PBC), and Environmental Consciousness (EC) would influence the Environmentally Responsible Purchase Intention (ERPI), which are discussed as follows.

First, Attitudes (ATT) determined Environmentally Responsible Purchase Intention (ERPI) during the COVID-19 pandemic. Our evidence is aligned with previous research indicating that behavioral attitudes are an essential factor in purchasing process when products are created based on eco-friendly methods that facilitate further recycling behavior [22]. Based on the above, this research is aligned with previous research by arguing that ERPI is influenced by Attitudes [55,56,57]. This is consistent with Shen et al. [25], who have argued that during COVID-19, the attitude toward the intention to purchase products has increased. One possible explanation for this outcome is that quarantines and voluntary self-isolation during the have had an emotional impact on consumers, which is in line with the perspective of Nguyen et al. [121] and the relation of ATT and consumer behavior, specifically in the youngest generations. For example, it has been highlighted in Colombia Attitude is one of the most critical aspects of decision-making from social and economic perspectives [122].

Second, our research also suggests that COVID-19 determined Perceived Behavioral Control (PBC). This conclusion is consistent with prior studies such as Xu et al. [23], Lucarelli et al. [71], and Vu et al. [56], who found that PBC affects purchasing intention related to pro-environmental behavior. Mexico and Chile support the hypothesis, both countries being Latin American members of the OECD. It last could partially explain political and economic conditions that are necessary to be part of this international organization and that directly influence how those countries promote economic development, but that consider elements of protection of the environment, which could be reflected in the behavior of consumers. In terms of age, this hypothesis was supported in generations Y and Z, reflecting that COVID-19 would contribute to the relationship between Perceived Behavioral Control (PBC) and Environmentally Responsible Purchase Intention (ERPI) in the younger generations.

Third, our evidence indicates that COVID-19 determined Environmental Consciousness (EC) in Latin American countries. In line with previous studies, our results show that Environmental Consciousness (EC) is an essential motivator for developing behavioral intention [1,41]. It is important to note that the results show that the pandemic scenario positively influenced the relationship between Environmental Consciousness (EC) and Environmentally Responsible Purchase Intention (ERPI) in all countries and all generations, demonstrating that this worldwide crisis has generated an effect on consumers, which could demonstrate the starting point towards a transition that establishes environmentally friendly behaviors [5,75]

On the opposite side, the current investigation rejects H2. In this sense, our results have shown that the COVID-19 pandemic has not positively contributed to the relationship between Subjective Norms (SN) and Environmentally Responsible Purchase Intention (ERPI). This can be explained because quarantines and social distancing have significantly reduced interaction between family, friends, and co-workers. Consequently, there were not enough social encounters for a social influence on behaviors toward an environmentally responsible purchase intention.

### 5.1. Practical and Managerial Implications

Different from the existing research in a context without a COVID-19 pandemic on the factors of Environmental Responsibly Purchase intention (ERPI) [15,23,51,52], our research proposes a research framework in Latin America focused on the effects of the COVID-19 pandemic, specifically on consumer behavior regarding Attitude (ATT), Subjective Norm (SN), Perceived Behavioral Control (PBC), and Environmental Consciousness (EC).

Regarding the practical implications of this research, there is a significant finding about the moderating effect on Environmentally Responsible Purchase Intention across generational groups. Our results indicate that age plays a moderator role in Environmentally Responsible Purchase Intention (ERPI) in generation X. Moreover, Perceived Behavioral Control (PBC) influences the younger groups (Y and Z). In this sense, young adults (Generation Z) and adults (Generation Y) are the ones who are susceptible to the potential impacts of COVID-19 in their intention to make an environmentally responsible purchase.

This could be explained due to younger groups manifesting greater environmental awareness [4] and online digitization [123], which has influenced the accessibility of information and environmental concerns during COVID-19. Building upon Nguyen [121], it is possible to argue that young consumers represent a powerful force in developing environmental awareness among the population, specifically within emerging markets such as the ones observed in Latin American countries. In this scenario, as a guideline for companies operating in this context, they should diversify their strategies to promote environmentally responsible purchase behavior by considering how different generations address the need for sustainability in purchasing processes, understanding that there is a different understanding and awareness of the environmental urgency on the planet.

This research offers suggestions and implications for managerial decision making. Our findings provide strong evidence of environmental developments in Latin American countries during the pandemic of COVID-19, allowing us to propose realistic prepositions for governments and companies. For example, Environmental Consciousness has increased significantly during the COVID-19 pandemic, manifested in consumers making efforts to prefer green products that are less harmful to the environment [4]. Therefore, companies should adopt sustainable and eco-friendly business practices to align with customers’ individual behavior. For example, businesses should promote the recycling and disposal of supplies in their packaging and stores to educate the population and facilitate the process [7]. In turn, advertising campaigns raise awareness of a brand that is friendly and close to its clients. In addition, business owners and executives must test new business models in response to the rise of online purchases and buy withdrawals at the collecting point. Since the quarantine and isolation during the COVID-19 pandemic, customers have temporarily or permanently altered their shopping habits.

Finally, local governments should evaluate this social context and support activities that positively influence our world from a national approach. The authorities should develop policies that help the transformation towards a more sustainable world—for example, proposing public programs to reduce pollution, recycling incentives, and promoting efficient water consumption.

### 5.2. Limitations and Future Research

This research has addressed the COVID-19 effects on environmentally responsible behavior, delivering theoretical and practical implications. However, some limitations should be considered for future research.

The first limitation refers to the sample; the sample is a non-probabilistic and simple cross-section. The participants responded voluntarily, giving their perception of the environmentally responsible purchase Intention. Additionally, this research only included participants from Chile, Colombia, Mexico, and Peru.

For this reason, it is recommended in future research to expand the sample to other Latin American countries, such as Argentina, Brazil, Bolivia, Uruguay, and Venezuela in South America; or Central American countries, such as Panama, Puerto Rico, the Dominican Republic, or Costa Rica [124,125]. As a result, a more extensive and diverse sample is expected to provide an opportunity to develop a cross-cultural study [126,127] or theoretical model based on human values in the context of geographical distances [128]. However, this limitation makes it difficult to generalize the results obtained in this scenario. Consequently, it is thus advised an expansion of this study employing a stratified random sample to compare generation consumers or cross-country consumer research [129]. In this context, the sample distribution should be extended for different age groups to analyze the possible effect size.

Secondly, this study only focused on the middle of the COVID-19 pandemic (between the third and fourth waves) in consumer behavior in the context of environmentally responsible purchase intention. For this reason, our results are not generalizable to the post-pandemic stage. Future studies may consider the environmental situation in a post-pandemic context to identify if the eco-friendly and pro-environmental behavior is maintained over time or was just a stationary effect.

Thirdly, another limitation is the method of collecting data. Our research carried out an online survey like most studies in times of COVID-19 due to quarantines, limited capacity, and social distancing. Therefore, consumer behavior has not been evaluated in real-time, so future post-pandemic research could use field surveys to immediately consult consumers about their purchase intention and behavior.

Finally, a fourth relevant limitation is people’s honesty when responding. As in other studies on pro-environmental behavior, it is possible that the respondents feel a social and ethical pressure to respond to show environmental interest. For this reason, the results must be evaluated and generalized with caution. Consequently, it is crucial to understand the gap between purchase intention and purchase experience. This occurs because a group of consumers provides a positive attitude toward society about a friendly purchase with the environment. However, these do not usually finalize the purchase. For this reason, future research should focus on filter questions to identify an effective and real purchase intention.

## Figures and Tables

**Figure 1 ijerph-20-03330-f001:**
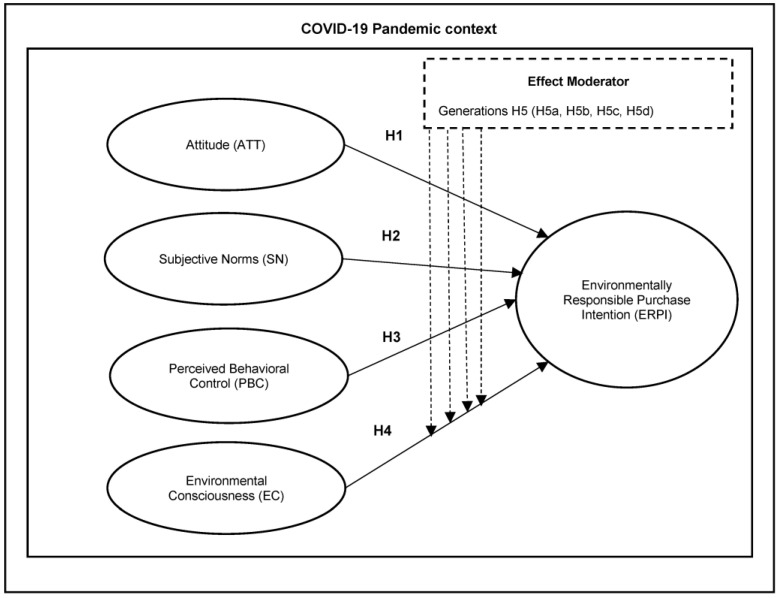
Illustration of research model hypotheses.

**Figure 2 ijerph-20-03330-f002:**
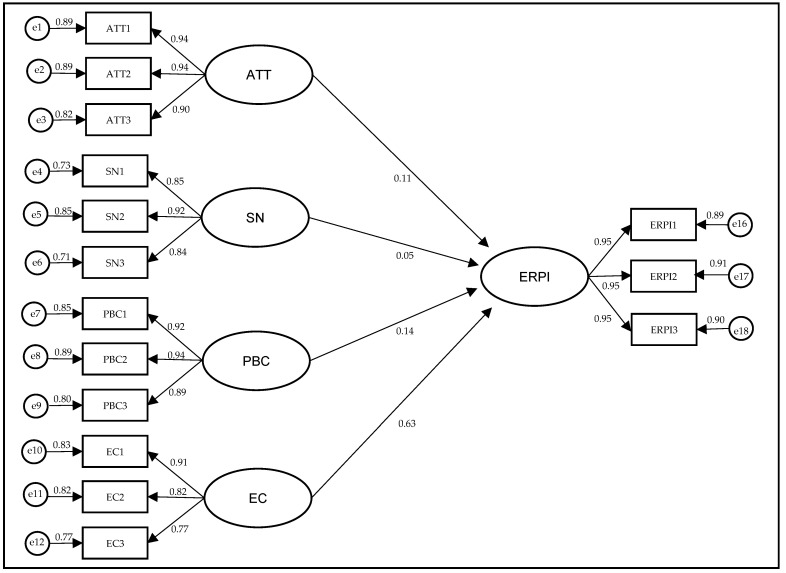
Structural model.

**Table 1 ijerph-20-03330-t001:** Potency analysis.

	Potency ^b^	Testing Assumptions				
		N1	N2	Standard Deviation ^c^	Effect Size	Sig.
Test for mean difference	0.980	197	793	1.25	0.320	0.05

^b^ It is based on the noncentral t-distribution. ^c^ Group variances are assumed equal.

**Table 2 ijerph-20-03330-t002:** Constructs and items.

Dimension	Item	Resource
Attitude (ATT)	(ATT1) The COVID-19 pandemic has caused me usually to favor buying products that make use of eco-friendly material.	[15,22]
	(ATT2) The COVID-19 pandemic has caused me to usually buy products that can be recycled.	
	(ATT3) The COVID-19 pandemic has caused me to buy eco-friendly products, even if they are not from a well-known company.	
Subjective Norms (SN)	(SN1) The COVID-19 pandemic has caused the people I listen to could influence me to purchase organic products.	[15,100]
	(SN2) The COVID-19 pandemic has caused people important to me think I should purchase eco-friendly products.	
	(SN3) The COVID-19 pandemic has caused my family and friends to think purchasing eco-friendly products is good.	
Perceived Behavioral Control (PBC)	(PBC1) The COVID-19 pandemic has caused me to always try to purchase environmentally responsible products.	[15,101]
	(PBC2) The COVID-19 pandemic has caused me to be confident I will purchase eco-friendly products when I go to buy.	
	(PBC3) The COVID-19 pandemic has caused me to have the resources and opportunities to buy eco-friendly products.	
Environmental Consciousness (EC)	(EC1) The COVID-19 pandemic has caused me willing to make some exceptional attempts to purchase eco-friendly products to protect the environment.	[15,102,103]
	(EC2) The COVID-19 pandemic has caused me to change my product brands due to ecological reasons.	
	(EC3) The COVID-19 pandemic has caused me when I have a choice will purchase a product that is less harmful to the environment.	
Environmentally Responsible Purchase Intention (ERPI)	(ERPI1) The COVID-19 pandemic has caused me to plan to buy ecological products in future.	[15]
	(ERPI2) The COVID-19 pandemic has caused me to plan to buy ecological products regularly.	
	(ERPI3) The COVID-19 pandemic has caused me to expend more effort on ecological products than on traditional ones.	

Source: Self-elaboration.

**Table 3 ijerph-20-03330-t003:** Scale items, factor loadings, composite reliabilities, and average variance extracted.

Constructs	Items	SD	Loading Factors	Regression Weights	KMO	CA	CR	AVE
Attitude (ATT)	ATT1	1.279	0.958	0.945	0.770	0.951	0.952	0.868
	ATT2	1.29	0.961	0.944				
	ATT3	1.277	0.944	0.905				
Subjective Norm (SN)	SN1	1.234	0.906	0.855	0.729	0.904	0.908	0.766
	SN2	1.257	0.942	0.924				
	SN3	1.267	0.900	0.845				
Perceived Behavioral Control (PBC)	PBC1	1.294	0.945	0.919	0.765	0.941	0.942	0.843
	PBC2	1.275	0.956	0.941				
	PBC3	1.273	0.937	0.894				
Environmental Consciousness (EC)	EC1	1.262	0.944	0.916	0.760	0.926	0.926	0.808
	EC2	1.286	0.932	0.907				
	EC3	1.295	0.925	0.873				
Environmental Responsibly Purchase intention (ERPI)	ERPI1	1.289	0.965	0.945	0.782	0.965	0.965	0.901
	ERPI2	1.288	0.969	0.953				
	ERPI3	1.294	0.966	0.949				

Source: Self-elaboration. Note: SD = standard deviation; KMO = Kaiser–Meyer–Olkin; CA = Cronbach’s alpha; CR = Composite Reliability; AVE = Average Variance Extracted.

**Table 4 ijerph-20-03330-t004:** Fornell Larcker criteria for discriminant validity.

Variables	ATT	SN	PBC	EC	ERPI
Attitude (ATT)	0.932				
Subjective Norm (SN)	0.807	0.875			
Perceived Behavioral Control (PBC)	0.848	0.85	0.918		
Environmental Consciousness (EC)	0.807	0.781	0.852	0.899	
Environmental Responsibly Purchase intention (ERPI)	0.796	0.767	0.832	0.895	0.949

Source: Self-elaboration.

**Table 5 ijerph-20-03330-t005:** Heterotrait–Monotrait ratio for discriminant validity.

Variables	ATT	SN	PBC	EC	ERPI
Attitude (ATT)					
Subjective Norm (SN)	0.807				
Perceived Behavioral Control (PBC)	0.848	0.850			
Environmental Consciousness (EC)	0.808	0.781	0.852		
Environmental Responsibly Purchase intention (ERPI)	0.797	0.768	0.832	0.895	

Source: Self-elaboration.

**Table 6 ijerph-20-03330-t006:** Model comparisons for CFA Models with marker variable.

Model	$X^{2}$	(df)	CFI	RMSEA	$\mathrm{LR}\;\mathrm{of}\;\Delta X^{2}$
CFA with marker	536.8	126	0.987	0.045	
Baseline	752.4	131	0.98	0.054	
Method-Constrained	568.4	130	0.986	0.046	184, *ρ* = 0.00000000
Method-Unconstrained	509.1	116	0.988	0.046	59.3, *ρ* = 0.00002834
Method-Restricted	521	126	0.988	0.044	9.52, *ρ* = 0.484

Notes: CFA = Confirmatory factor analysis; RMSEA: Root mean square error of approximation; LR: Likelihood ratio test.

**Table 7 ijerph-20-03330-t007:** SEM estimations on hypothesis tests.

Hypothesis	Structural Path	Estimate	*p*-Value	SE	CR	Decision
H1	ATT → ERPI	0.117	***	0.029	4.043	Supported
H2	SN → ERPI	0.053	0.078	0.034	1.708	Not supported
H3	PBC →ERPI	0.147	***	0.038	3.921	Supported
H4	EC → ERPI	0.635	***	0.033	20.099	Supported

Source: Self-elaboration. Note: AT = Attitude; SN = Subjective Norms; PBC = Perceived Behavioral Control; EC = Environmental Consciousness; ERPI = Environmental Responsibly Purchase intention; SD = standard deviation; CR = Composite Reliability; *p*-value = *** *p* < 0.01.

**Table 8 ijerph-20-03330-t008:** Moderator effect of the country and generations by multi-group analysis.

	Variable	H	SP	E	*p*-Value	SE	CR	Decision
Generation	Generation X	H1	ATT → ERPI	−0.019	0.803	0.081	−0.25	Not supported
		H2	SN → ERPI	−0.013	0.850	0.082	−0.19	Not supported
		H3	PBC → ERPI	0.113	0.300	0.120	1.305	Not supported
		H4	EC → ERPI	0.663	***	0.090	7.602	Supported
	Generation Y	H1	ATT → ERPI	0.076	0.160	0.054	1.406	Not supported
		H2	SN → ERPI	0.040	0.435	0.059	0.781	Not supported
		H3	PBC → ERPI	0.124	0.047 **	0.065	1.99	Supported
		H4	EC → ERPI	0.754	***	0.082	9.574	Supported
	Generation Z	H1	ATT → ERPI	0.164	***	0.040	4.144	Supported
		H2	SN → ERPI	0.047	0.287	0.050	1.065	Not supported
		H3	PBC → ERPI	0.191	***	0.054	3.558	Supported
		H4	EC → ERPI	0.502	***	0.054	10.02	Supported

Source: Self-elaboration. Note: H = Hypothesis; SP = Structural Path; E = Estimate; AT = Attitude; SN = Subjective Norms; PBC = Perceived Behavioral Control; EC = Environmental Consciousness; ERPI = Environmental Responsibly Purchase intention *p*-value = *** *p* < 0.01 ** *p* < 0.05; SE = standard error; CR = Composite Reliability.

**Table 9 ijerph-20-03330-t009:** Invariance.

Invariance	Overall Model	$\chi^{2}$	Df	*p*-Value	Invariant
Invariance generation	Unconstrained	981.7	242		
	Fully constrained	1005.8	263		
	Difference	23.924	21		
	Number of groups	2		0.297	YES
	Critical Chi square	29.615			

Source: Self-elaboration.

## Data Availability

Data availability can be requested by writing to the corresponding author of this publication.

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
