# Peer review of "COVID-19 Effects on Environmentally Responsible Behavior: A Social Impact Perspective from Latin American Countries"

_ijerph, 2023, doi:10.3390/ijerph20043330_

Round 1

Reviewer 1 Report (Previous Reviewer 2)

I enjoyed reading the content of the article. The topic is interesting and current. Here are my comments and suggestions:

Keywords

The keywords are too long. The rule is that keywords should consist of one word or two in a phrase. The more words in the phrase, the harder it is to search for an article by keywords.

Introduction

a/ The introduction lacks a statement of the purpose of the research. The authors only mention what they intend to do, but that is not the same as the goal.

b/ The authors should define the research gap more clearly - in this form it is ambiguous, poorly expressed.

c/ The Theory of Planned Behavior should be explained here - what is the essence of this theory, who, when and in what circumstances formulated it, etc.

Theoretical background

a/ The beginning theoretical background is not good; is chaotic - it needs to be rethought.

b/ Is the caption under figure 1 "Framework e integrated model." it is correct?

c/ At what time should H1 be written - past or present tense?

d/ Proposes H2 to read as follows: Subjective Standards (SN) determined Consumers' Environmentally Conscious Purchasing Intent (ERPI) during COVID-19.

e/ Suggests that H2 read as follows: Perceived Behavioral Control (PBC) influenced consumers' Environmentally Conscious Purchasing Intent (ERPI) during COVID-19.

f/ H45 - ?

g/ Proposes H4 to read as follows: Environmental Awareness (EC) determined Consumers' Environmentally Conscious Purchasing Intent (ERPI) during COVID-19.

Results

a/ Is Figure 2 included in its entirety? Conclusions a/ There is no reference to hypotheses.

b/ There are no specific conclusions.

c/ What new did this research bring to the theory? What empirical implications did the study make? What recommendations for managers can be formulated based on the research results?

d/ Directions for future research should be at the end of the conclusions. Also, this thread needs to be expanded.

In conclusion, the article still needs to be rethought and corrected. I also recommend that you pay attention to the fact that the English language is better.

Author Response

Dear reviewer 1:

First, we want to thank you for allowing us to revise and resubmit our paper called “COVID-19 effects on environmentally responsible behavior: a social impact perspective from Latin American countries”. Manuscript ID: ijerph-2145566).

We have carefully considered all the comments raised by reviewers. Below, we explain in more detail how our paper's new version addresses your suggestion raised.

The manuscript has been edited using the Journal template and format. To facilitate the evaluation of the changes, we work with "Track Changes"

Hopefully, you will agree that our changes have resulted in a stronger paper, and we thank you for the constructive and helpful comments.

Yours sincerely

The authors

Reviewer 2 Report (New Reviewer)

1. From the title, we can see that the theme of the paper is related to COVID-19. However, there is no variable related to COVID-19 in the research model. There is inconsistency between the two.

2. In the introduction, the environmental problems caused by COVID-19 should be strengthened. At the same time, it should clearly highlight what the research problem is.

3. In Theoretical background, the previous research on Environmentally Responsible Purchase Intention should be added. At the same time, the author should explain why choose environmental awareness rather than other variables.

4. Further enrich managerial implications.  

5. Add the latest literature, such as 2022

Author Response

Dear reviewer 2:

First, we want to thank you for allowing us to revise and resubmit our paper called “COVID-19 effects on environmentally responsible behavior: a social impact perspective from Latin American countries”.(Manuscript ID: ijerph-2145566).

We have carefully considered all the comments raised by reviewers. Below, we explain in more detail how our paper's new version addresses your suggestion raised.

The manuscript has been edited using the Journal template and format. To facilitate the evaluation of the changes, we work with "Track Changes"

Hopefully, you will agree that our changes have resulted in a stronger paper, and we thank you for the constructive and helpful comments.

Yours sincerely

The authors

Round 2

Reviewer 1 Report (Previous Reviewer 2)

The article has been completely remodeled. The authors took into account most of my comments. There are, of course, still debatable issues, but they are not important enough to negatively evaluate the article. I can recommend an article for publication.

This manuscript is a resubmission of an earlier submission. The following is a list of the peer review reports and author responses from that submission.

Round 1

Reviewer 1 Report

I have now had the opportunity to read and review the manuscript "COVID-19 effects on environmentally responsible behavior: a social impact perspective from Latin American countries." submitted to IJERPH.

This paper argues that an extended TPB model could contribute to the explanation of COVID19 related effects/experiences on responsible consumption. As such, the paper is largely well-written (though some language editing would be necessary) and the latin-american samples are a strong contribution to the primarily "westernized" samples. Similarly, the SEM invariance Analysis are state-of-the-art (but see below) and I believe this paper could make an important contribution.

However, there are various conceptual issues that I believe need thorough consideration.

1) Most importantly, I am not sure that the extension of TPB as proposed makes sense. Why would "WTP" be a comparable and same-level predictor as attitudes, perceived behavioral control and subjective norms? All these are based on strog psychological theorizing and represent complex constructs (see also 2nd comment). WTP, however, is more of result of these constructs so that I do not see any reason why it should be in the same line with the others. Rather, I would think that WTP is a consequence of intention; or just another formalization of intention.

2) The concepts used are underdeveloped. Attitude is not just "positive or negative valence", it is based on expectany x value considerations so that I would build this up more succinctly. For norms and PBC; the description is sufficient I believe.

3) I would somewhat tone down the conceptual novelty of the corona - environment link. While I think the empirical work of the authors is innovative, there is already other research dealing with the corona - environment nexus, building on previous models of pro-environmental behavior (see for example the work by Hamann, Tröger etc., 2020, Journal of Environmental Psychology, https://doi.org/10.1016/j.jenvp.2020.101444). I think it would be valuable to focus more strongly on such existing models, also with regard to WTP which is a pro-environmental response, based on various psychological processes (including collective ones).

3) As for the hypotheses, most of them follow from the conceptual analysis, but the moderating (2.2) hypotheses are unclear - what is the theoretical rationale for these? And why should this be relevant? I could not find any indication for this in the theoretical introduction.

4) Looking at the items on page 8, it becomes again evident that WTP could be seen as same level variable as attitude, norms and PBC. I would highly suggest to treat WTP as what it is: A consequence of psychological constructs, something that is more of an intention. Just screening google scholar with the search terms "willingness to pay" and "intentions" shows that it is treated as outcome, not as predictor.

5) How did the authors statistically control for the extremely different sub sample sizes regarding "generation"? This is relevant to most analyses, most importantly however with regard to the invariance testing that has relatively strict conditions. (see e.g. Davidov et al., 2012; Steenkamp & Baumgartner, 1998).

6) At the beginning of the results section, it remains unclear whether the parameters reported refer to the whole sample (across countries) or within countries. Indeed, I miss a lot of information right from the start dealing with the nature of the samples.

7) Given that there was not theoretical rationale for the Moderator effects, I was little surprised that there were hardly relevant effects. I highly suggest to rethink the role of both moderators and delineate why they should have an effect in the first place.

8) In the discussion (and everywhere else), please refrain from causal language. Nothing has an impact here on anything, it is all correlational and related to each other.

9) In lines 439, the authors write that COVId19 positively impacted EC - how did you find out? This is no longitudinal pre-post design. I would really be careful with such wording that communicates more into the study and data than there actually is. (check for this throughout). In fact, I would be very careful interpreting this as "covid19 led people to...", as the whole design does not allow such a test. It is just showing whether people think it affected their attitudes etc, but a longitudinal design (difficult to pre-covid data now...) would be the way to go.

Author Response

Dear reviewer 1:

First, we want to thank you for allowing us to revise and resubmit our paper called “COVID-19 effects on environmentally responsible behavior: a social impact perspective from Latin American countries”.(Manuscrip ID: ijerph-1960381).

We have carefully considered all the comments raised by reviewers. Below, we explain in more detail how our paper's new version addresses your suggestion raised.

The manuscript has been edited using the Journal template and format. To facilitate the evaluation of the changes, we work highlighting the text in light blue

Hopefully, you will agree that our changes have resulted in a stronger paper, and we thank you for the constructive and helpful comments.

Yours sincerely

The authors

Reviewer 2 Report

I am pleased to read the content of the article. The topic is interesting and up-to-date, new. Below I present my comments and suggestions:

Abstract The aim of the research is not clearly defined and the aim is an integral part of the abstract. He proposes to write directly: "The aim of the research is ..." Introduction There is also no clear purpose in the introduction. However, we know what the Authors want to do - because they write about it. The goal is shown only in the methodology. But a scientific article has its own rules and should contain these basic elements.

The literature review is thoughtful. In relation to the subject, it is good.

The methodology is well described and justified. The discussion is impressive. It is a strong part of the article.

Conclusions - the authors did not indicate the limitations of the research.

Author Response

Dear reviewer 2:

First, we want to thank you for allowing us to revise and resubmit our paper called “COVID-19 effects on environmentally responsible behavior: a social impact perspective from Latin American countries”.(Manuscrip ID: ijerph-1960381).

We have carefully considered all the comments raised by reviewers. Below, we explain in more detail how our paper's new version addresses your suggestion raised.

The manuscript has been edited using the Journal template and format. To facilitate the evaluation of the changes, we work highlighting the text in light blue

Hopefully, you will agree that our changes have resulted in a stronger paper, and we thank you for the constructive and helpful comments.

Yours sincerely

The authors

Round 2

Reviewer 1 Report

I have now had the opportunity to read the revision of the paper.

While I see some effort in improving the paper, I am still not convinced that this work is conceptually sound, especially with regard to the role of WTP as an equal predictor to the TPB variables. I do not see any additional theoretical or conceptual effort to reconcile these previous comment.

Also, there are still papers on the covid - PEB link not included so that the paper suggests innovation where there is none.

Maybe other reviewers come to a different conclusion or recommendation but from a theory-based environmental psychology perspective, I do not recommend acceptance of this manuscript.